# Topography of cancer-associated immune cells in human solid tumors

**Jakob Nikolas Kather**[1,2,3,4]*, **Meggy Suarez-Carmona**[1,3],
**Pornpimol Charoentong**[1,3], **Cleo-Aron Weis**[5], **Daniela Hirsch**[5], **Peter Bankhead**[6],
**Marcel Horning**[1], **Dyke Ferber**[1,3], **Ivan Kel**[1], **Esther Herpel**[7,8], **Sarah Schott**[9],
**Inka Zörnig**[1,3], **Jochen Utikal**[10,11], **Alexander Marx**[5], **Timo Gaiser**[5],
**Herrmann Brenner**[2,12,13], **Jenny Chang-Claude**[14,15], **Michael Hoffmeister**[12],
**Dirk Jäger**[1,2,3], **Niels Halama**[1,2,3]*

[1]Department of Medical Oncology and Internal Medicine VI, National Center for Tumor Diseases, University Hospital Heidelberg, Heidelberg, Germany; [2]German Cancer Consortium, Heidelberg, Germany; [3]Applied Tumor Immunity, German Cancer Research Center, Heidelberg, Germany; [4]Division of Gastroenterology, Hepatology and Hepatobiliary Oncology, University Hospital RWTH Aachen, Aachen, Germany; [5]Department of Pathology, University Medical Center Mannheim, Heidelberg University, Mannheim, Germany; [6]Northern Ireland Molecular Pathology Laboratory, Centre for Cancer Research and Cell Biology, Queen's University Belfast, Northern Ireland, United Kingdom; [7]Institute of Pathology, University Hospital Heidelberg, Heidelberg, Germany; [8]Tissue Bank of the National Center for Tumor Diseases, Heidelberg, Germany; [9]Department of Gynecology, University Hospital Heidelberg, Heidelberg, Germany; [10]Skin Cancer Unit, German Cancer Research Center, Heidelberg, Germany; [11]Department of Dermatology, Venereology and Allergology, University Medical Center Mannheim, Heidelberg University, Mannheim, Germany; [12]Division of Clinical Epidemiology and Aging Research, German Cancer Research Center, Heidelberg, Germany; [13]Division of Preventive Oncology, German Cancer Research Center and National Center for Tumor Diseases, Heidelberg, Germany; [14]Division of Cancer Epidemiology, German Cancer Research Centre, Heidelberg, Germany; [15]Cancer Epidemiology Group, University Cancer Center Hamburg, University Medical Center Hamburg-Eppendorf, Hamburg, Germany

**\*For correspondence:**
jakob.kather@nct-heidelberg.de (JNK);
niels.halama@nct-heidelberg.de (NH)

**Competing interests:** The authors declare that no competing interests exist.

**Abstract** Lymphoid and myeloid cells are abundant in the tumor microenvironment, can be quantified by immunohistochemistry and shape the disease course of human solid tumors. Yet, there is no comprehensive understanding of spatial immune infiltration patterns ('topography') across cancer entities and across various immune cell types. In this study, we systematically measure the topography of multiple immune cell types in 965 histological tissue slides from N = 177 patients in a pan-cancer cohort. We provide a definition of inflamed ('hot'), non-inflamed ('cold') and immune excluded patterns and investigate how these patterns differ between immune cell types and between cancer types. In an independent cohort of N = 287 colorectal cancer patients, we show that hot, cold and excluded topographies for effector lymphocytes (CD8) and tumor-associated macrophages (CD163) alone are not prognostic, but that a bivariate classification system can stratify patients. Our study adds evidence to consider immune topographies as biomarkers for patients with solid tumors.
DOI: https://doi.org/10.7554/eLife.36967.001

## Introduction

Malignant tumors growing in an immunocompetent host elicit an immune response, evident by the presence of various inflammatory/immune cell in tumor tissue (*Shalapour and Karin, 2015*; *Mantovani et al., 2008*; *Bindea et al., 2013*). In order to grow to a clinically relevant size, tumor cells develop specific escape mechanisms against the immune system by manipulating inflammatory cells for their benefit (*de Visser et al., 2006*; *Dunn et al., 2002*; *Fridman et al., 2013*). One of the key strategies is that tumor cells interfere with immune signaling, hijacking immunosuppressive cells and thereby shaping the immune infiltrate, which allows for tumor cell proliferation (*Chen and Mellman, 2013*; *Chen and Mellman, 2017*).

These mechanisms have been in the focus of oncology for several years (*Kather et al., 2018a*). Currently a number of immunotherapeutic drugs are available which interfere with immune cells in the tumor microenvironment in order to facilitate tumor control (*Becht et al., 2016a*; *Galluzzi et al., 2014*). However, the complex nature of immune infiltrates impairs the development of more targeted approaches. Specifically, tailored combination treatments are widely proposed as a way to more effective cancer therapy (*Sharma and Allison, 2015a*; *Sharma and Allison, 2015b*; *Zitvogel et al., 2011*). Systematically deciphering tumor-immune phenotypes is key to a better understanding and more effective tailoring of immunotherapies (*Greenplate et al., 2016*).

Analysis of solid tumor tissue slides by immunohistochemistry (IHC) is the gold standard to assess tumor immune infiltrate because it allows for exact quantification of type, density and localization of immune cells (*Fridman et al., 2017*; *Becht et al., 2016b*). For more than a decade, digital pathology has been the method of choice to reliably and reproducibly analyze large cohorts of patient samples and can provide potential biomarkers for immunotherapy (*Becht et al., 2016a*; *Kather et al., 2016*; *Gurcan et al., 2009*). Immune cell quantification in digitized tissue has been used to identify robust and clinically relevant biomarkers in numerous cancer entities, for example in colorectal cancer (CRC) primary tumors and liver metastases (*Galon et al., 2006*; *Halama et al., 2011*; *Mlecnik et al., 2016*). Histological analysis of tumor-infiltrating lymphoid cells has been proven to be a reliable and prognostically relevant marker (*Galon et al., 2014*; *Denkert et al., 2016*; *International TILs Working Group 2014 et al., 2015*). Antitumor immunity arises in a complex ecosystem of various cell types that closely interact with one another, such as effector lymphocytes (*Li et al., 2016*), macrophages (*Halama et al., 2016*; *Biswas and Mantovani, 2010*), dendritic cells (*Gardner and Ruffell, 2016*), granulocytes (*Coffelt et al., 2016*), innate lymphoid cells (*Crome et al., 2017*), regulatory T cells (*Nishikawa and Sakaguchi, 2010*), natural killer cells (*Crome et al., 2013*; *Barrow and Colonna, 2017*), myeloid-derived suppressor cells (*Talmadge and Gabrilovich, 2013*) and other cell types (*Kather et al., 2017*). In tumor tissue, T lymphocytes (T cells) and macrophages are among the most abundant immune cells and are closely related to clinical outcome (*Fridman et al., 2017*; *Kather et al., 2017*; *Fridman et al., 2012*; *Kather et al., 2018b*).

After years of detailed IHC analysis of solid tumor slides, a paradigm for the classification of tumor-immune phenotypes has emerged and three classes of tumors are generally assumed: 'cold' tumors (or 'immune desert', showing no immune cell infiltration), 'immune-excluded' tumors (with immune cells aggregating at the tumor boundaries) and 'hot' tumors (or 'inflamed' tumors, showing pronounced immune infiltrates in the tumor core (*Chen and Mellman, 2017*; *Lanitis et al., 2017*; *Joyce and Fearon, 2015*). However, although this classification is generally accepted, it is backed by surprisingly little quantitative data. Fundamental biological questions regarding this concept are essentially unanswered, such as: Do these topographies exist in all tumor entities? Is there a difference between immunotherapy-sensitive and immunotherapy-insensitive tumor types? Does the concept of cold/excluded/hot apply to lymphoid and myeloid cells, or only to one of them? Do all tumors use the same strategies for immune escape?

In the last years, several large studies have systematically investigated immune-tumor phenotypes in detail. However, most of these studies were not suitable to distinguish cold, excluded and hot tumors because they did not look at the tumor core and the invasive margin at the same time. Two recent studies of immunophenotypes in colorectal cancer (CRC) have shown that the average immune cell density is higher around the tumor than in the tumor core (*Bindea et al., 2013*; *Mlecnik et al., 2016*). Yet, the concept of cold/excluded/hot tumors was not investigated in these

studies. Also, other comprehensive studies have not taken spatial patterns of immune cell phenotypes into account (*Becht et al., 2016*). Recent large-scale studies have looked at high-dimensional phenotypes of tumor-infiltrating immune cells, but have not specifically addressed different topographies (*Newell and Davis, 2014*; *Wong et al., 2016*; *Newell and Becht, 2018*; *Kather et al., 2018c*). Another previous study investigated spatial patterns of lymphoid and myeloid cells in human solid tumors – however, only tissue microarrays (TMA) were included in that study, precluding any possible differentiation between invasive margin and tumor core (*Tsujikawa et al., 2017*). Lastly, the correlation of the tumor immunophenotype to its transcription profile was investigated previously, but was lacking a spatially resolved approach (*Charoentong et al., 2017*).

In the present study, we have attempted to close this knowledge gap by a systematic analysis of a large cohort of various human tumors in a spatially resolved way. In particular, this included a systematic analysis of the following immune cells: CD3+ T-lymphocytes, CD8+ T-lymphocytes, PD1+ T lymphocytes, FOXP3+ T lymphocytes (which we assume to be largely 'regulatory', although the role of FOXP3 is more complex (*Wang et al., 2007*) and CD68+ and CD163+ monocytes/macrophages.

## Results

### Immune cell densities in major cancer entities

We measured immune cell densities (number of cells per mm$^2$, *Figure 1A–B*) in 965 immunostained histological tissue slides (listed in *Supplementary file 3*) in the pan-cancer cohort in three spatial compartments per slide (*Figure 1C*): outer invasive margin (0 – 500 µm outside the tumor invasion front), inner invasive margin (500 µm to the inside) and in the tumor core. In accordance with previous studies, colorectal adenocarcinoma primary tumors and liver metastases had a higher cell count in the outer invasive margin than in the tumor core (*Figure 2—figure supplement 2* and *Figure 2— figure supplement 3*). Melanoma samples had higher cell counts in the inner tumor than outside of the tumor for all analyzed cell types. The other tumor types showed less clear-cut patterns, highlighting the need for a more detailed analysis.

### Unsupervised clustering of lymphocyte densities separates hot and cold tumors

As previous studies have assumed the existence of 'hot', 'cold' and 'immune excluded' tumors with regard to lymphocyte infiltration, we assessed our data set for evidence of this clustering. We used cell densities for CD3+ and CD8+ cells in all three spatial compartments and used multiple methods to determine the optimal number of clusters. For CD3 and CD8 separately as well as for both together, the majority of the optimization runs converged on two clusters (*Figure 2—figure supplement 4*), mostly representing 'hot' and 'cold' tumors. There was no strong tendency to converge on three clusters, which means that there is no strong inherent grouping into 'hot', 'cold' and 'immune excluded' tumors.

### Conceptual definition of hot, cold and immune excluded tumors

We asked whether there is a rationale to define 'immune excluded tumors' as a separate group although it does not naturally emerge in unsupervised clustering methods. With three spatial compartments, each of them having either high or low cell density, there are in theory eight possible topographies. We asked whether this number could be reduced and analyzed the statistical correlation between the spatial compartments for each type of staining in all tumor types. We found that in general, there was a high correlation between 'tumor core' and 'inner invasive margin', but not between either compartment and the 'outer invasive margin' (*Figure 2—figure supplement 5*). 'Tumor core' and 'inner invasive margin' can be collapsed into one compartment because cell counts in these compartments are highly correlated. This leaves only four possible topographies that are by equivalent to three previously postulated phenotypes (*Chen and Mellman, 2017*): high density outside of the tumor with a low density inside the tumor can be described as 'immune excluded'. Low density inside and outside is 'cold' and high density inside the tumor is 'hot' regardless of cell density outside of the tumor (*Figure 2A–F*).

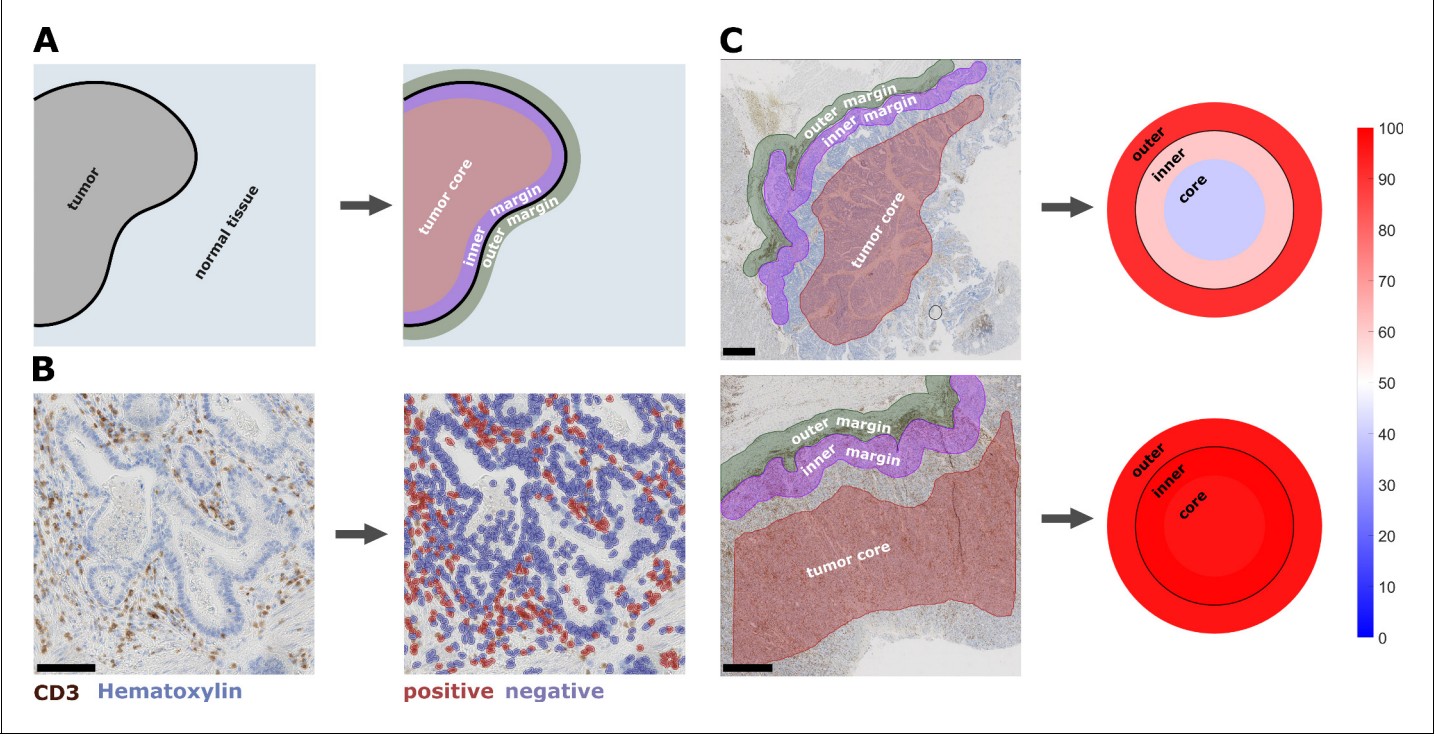

**Figure 1.** Semiautomatic image analysis defines immune cell topography. (**A**) Manual delineation of three compartments: outer 500 μm invasive margin, inner 500 μm invasive margin, tumor core. (**B**) Example of automatic cell detection in a CD3-stained gastric carcinoma slide. Left: original image, right: after cell detection and classification. (**C**) Cell counts in all three compartments can be used to create a 'target plot' (visualization resembling a shooting target) where the color of each compartment corresponds to the percentile-normalized cell density. Here, two examples of CD3-stained gastric carcinoma tissue slides are shown. The upper sample has an immune-excluded phenotype while the lower sample has an inflamed phenotype. Unit on the color scale: percentile-normalized cell density. Scale bar in B is 100 μm, scale bars in C are 1 mm.

DOI: https://doi.org/10.7554/eLife.36967.002

The following figure supplement is available for figure 1:

**Figure supplement 1.** Example images for cell count in all six immunostains.

DOI: https://doi.org/10.7554/eLife.36967.003

## Prevalence of immune topographies in different tumor types

Having operationalized these definitions of immune topographies, we next asked how they are distributed in different tumor types in the pan-cancer cohort. As a cutoff value for high versus low cell density we used the median cell density for each cell type (median number of cells per mm² in any tumor type in any compartment, listed in *Supplementary file 4*). Based on these definitions, different tumor types showed very different distributions of immune topographies (*Figure 3A–F*). Most strikingly, tumor types that are to some degree sensitive to approved cancer immunotherapies (such as melanoma (MEL), lung adeno (LUAD), lung squamous (LUSC) and head and neck squamous (HNSC) had a large proportion (approximately half) of CD3-hot, CD8-hot and PD1-hot tumors. In accordance with previous studies (*Halama et al., 2011*), colorectal primary (COAD-PRI) and colorectal metastatic (COAD-MET) had a very high proportion of CD3-excluded tumors (*Figure 3A*). Differences between tumor types were most pronounced for regulatory T-cells (Foxp3+ cells, *Figure 3D*) with more than half of lung adeno (LUAD), head and neck squamous (HNSC), stomach adenocarcinoma (STAD) and esophageal (ESCA) cancer samples having Foxp3-hot topographies. Also, while close to half of all analyzed COAD-PRI samples were Foxp3-hot, the vast majority of all COAD-MET samples were Foxp3-cold (*Figure 3A*).

## Bivariate classification of immune topographies

Subsequently, we asked whether the topography for a given immune cell type co-occurs with the same topography for other immune cell types – or whether tumors can be 'cold' for one immune cell

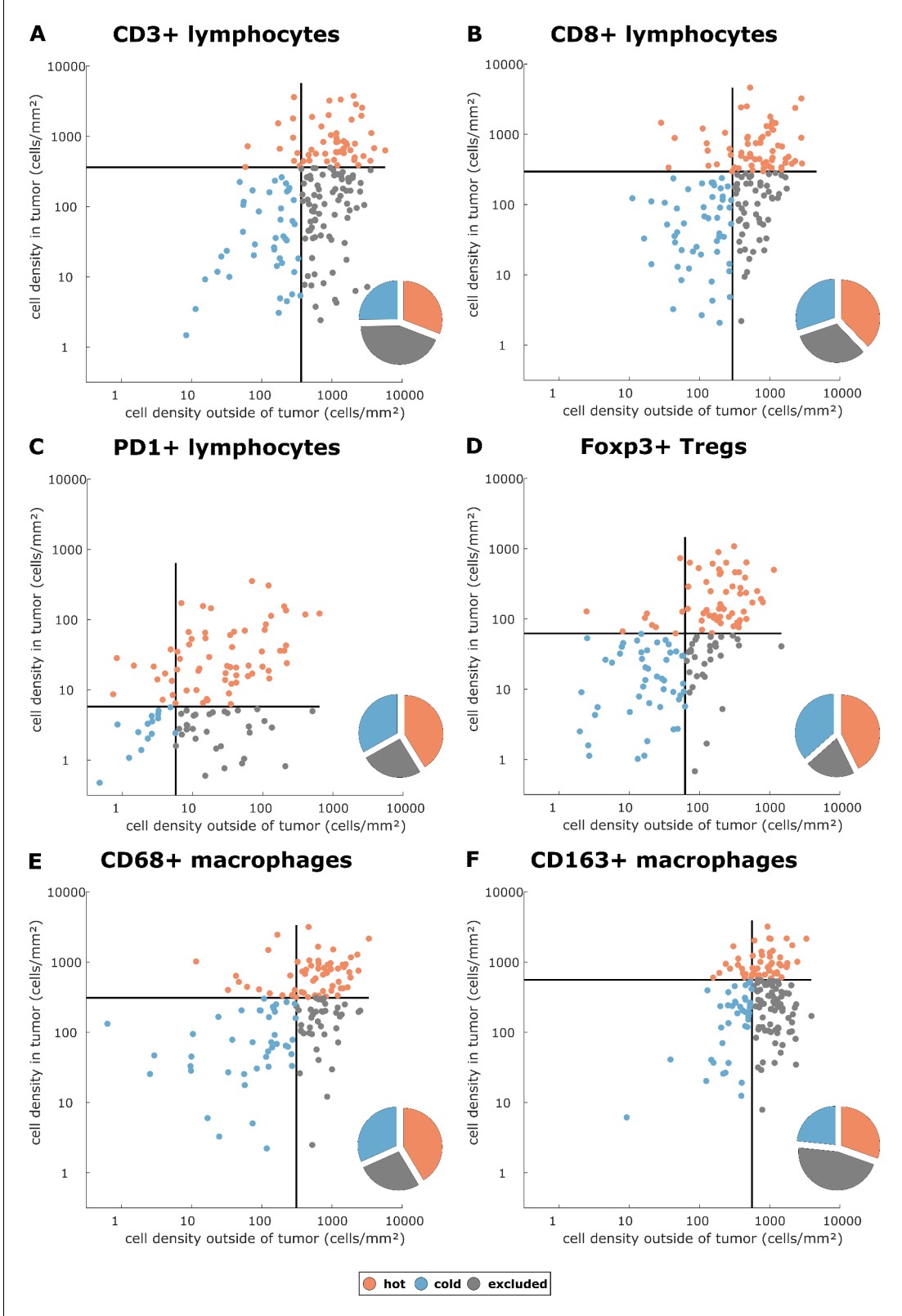

**Figure 2.** Cell densities in the tumor core and in the outer invasive margin in the pan-cancer cohort. Raw cell densities are plotted for each cell type and both major compartments. Gray lines indicate the median density for this cell type. Split at the median, tumors can be classified as cold, hot or immune excluded for all immune cell types. However, the scatter plot shows that for all immune cell types, there is no natural clustering into these phenotypes – the phenotypes blend into each other.

*Figure 2 continued on next page*

*Figure 2 continued*

DOI: https://doi.org/10.7554/eLife.36967.004

The following figure supplements are available for figure 2:

**Figure supplement 1.** Replication experiment for the full tissue analysis pipeline.

DOI: https://doi.org/10.7554/eLife.36967.005

**Figure supplement 2.** Normalized immune cell counts for cytotoxic T lymphocytes (CD8) and pro-tumor macrophages (CD163).

DOI: https://doi.org/10.7554/eLife.36967.006

**Figure supplement 3.** Average cell density percentile score for all compartments in ten tumor entities and six immunostains.

DOI: https://doi.org/10.7554/eLife.36967.007

**Figure supplement 4.** Optimal number of lymphocyte topography clusters arising in repeated optimization runs with different methods.

DOI: https://doi.org/10.7554/eLife.36967.008

**Figure supplement 5.** Correlations between cell densities in different spatial compartments.

DOI: https://doi.org/10.7554/eLife.36967.009

type and 'hot' or 'excluded' for another cell type at the same time. This question is important because in the clinic, immune topographies are often used as stratifying biomarkers for immunotherapy trials (*Kather et al., 2018c*) but it is unclear how many spatial compartments and which histological markers should be looked at.

Indeed, we saw that tumors often had different topographies for a pair of immune cell markers (*Figure 4*). Especially, regarding lymphocytic and myeloid markers, there was a discordance in a substantial number of cases that were lymphoid-hot and myeloid-cold (*Figure 4*). We further stratified this by different tumor entities (*Figure 5A–E*) and found pronounced differences between the tumor types. CD3 and CD8 topographies were mostly concordant (*Figure 5A*) as can be expected because CD8+ lymphocytes are a subset of CD3+ lymphocytes. Regarding CD8 vs. Tregs (*Figure 5B*), there was a notable discordance. For example, most colorectal primary tumors (COAD-PRI) that were Foxp3-excluded were CD8-cold. Analyzing tumor-associated macrophages (TAMs) labeled with CD68 (*Figure 5D*), we found that several melanomas (MEL) that were CD68-hot were CD8-cold which was rarely observed in other tumor types. Colorectal cancer liver metastases (COAD-MET) were mostly CD8-excluded, corresponding to a CD68-excluded or CD68-hot phenotype. This was similarly observed for CD163+ macrophages (*Figure 5E*). These findings suggest that tumors of different immunophenotypes require different ways of immune escape and that these mechanisms can only be distinguished when considering a two-dimensional myeloid-lymphoid classification system.

## Pan-cancer similarity based on spatial immune phenotype

Using a full panel of immune cell markers (CD3, CD8, PD1, Foxp3, CD68 and CD163), we asked how similar different tumor types were in terms of immune cell spatial layout. For all 144 tumors that were stained for this full panel (*Figure 6A–G*, six immunostains, three compartments, 144 patients, after percentile normalization for each cell type), we used unsupervised hierarchical clustering to define similarity. We found that ovarian cancer (OV) was an outlier (*Figure 6H*) due to its low infiltration with almost all immune cell subsets (*Figure 2—figure supplement 3*). Highly immunogenic tumor types such as melanoma (MEL), stomach cancer (STAD) and non-small cell lung cancer (NSCLC: LUAD and LUSC) were close in hierarchical clustering.

## Clinical utility of immune topography

Having shown that hot, cold and excluded immune patterns exist, and that these patterns vary between tumor types and immune cell types, we investigated the clinical utility of this classification system in an independent patient cohort (DACHS cohort). We analyzed the topography of CD8+ lymphocytes and CD163+ macrophages in colorectal cancer (CRC) primary tumors because these cell types have previously been linked to prognosis (*Fridman et al., 2012*) and showed discordant topographies in the pan-cancer cohort (COAD_PRI in *Figure 5E*).

We derived immune topographies for CD8 and CD163 from cell counts in the outer invasive margin and the tumor core for N = 287 patients using the median cell densities from the pan-cancer cohort as cutoff values (*Figure 7A and B*). As in the pan-cancer cohort, most patients were cold or

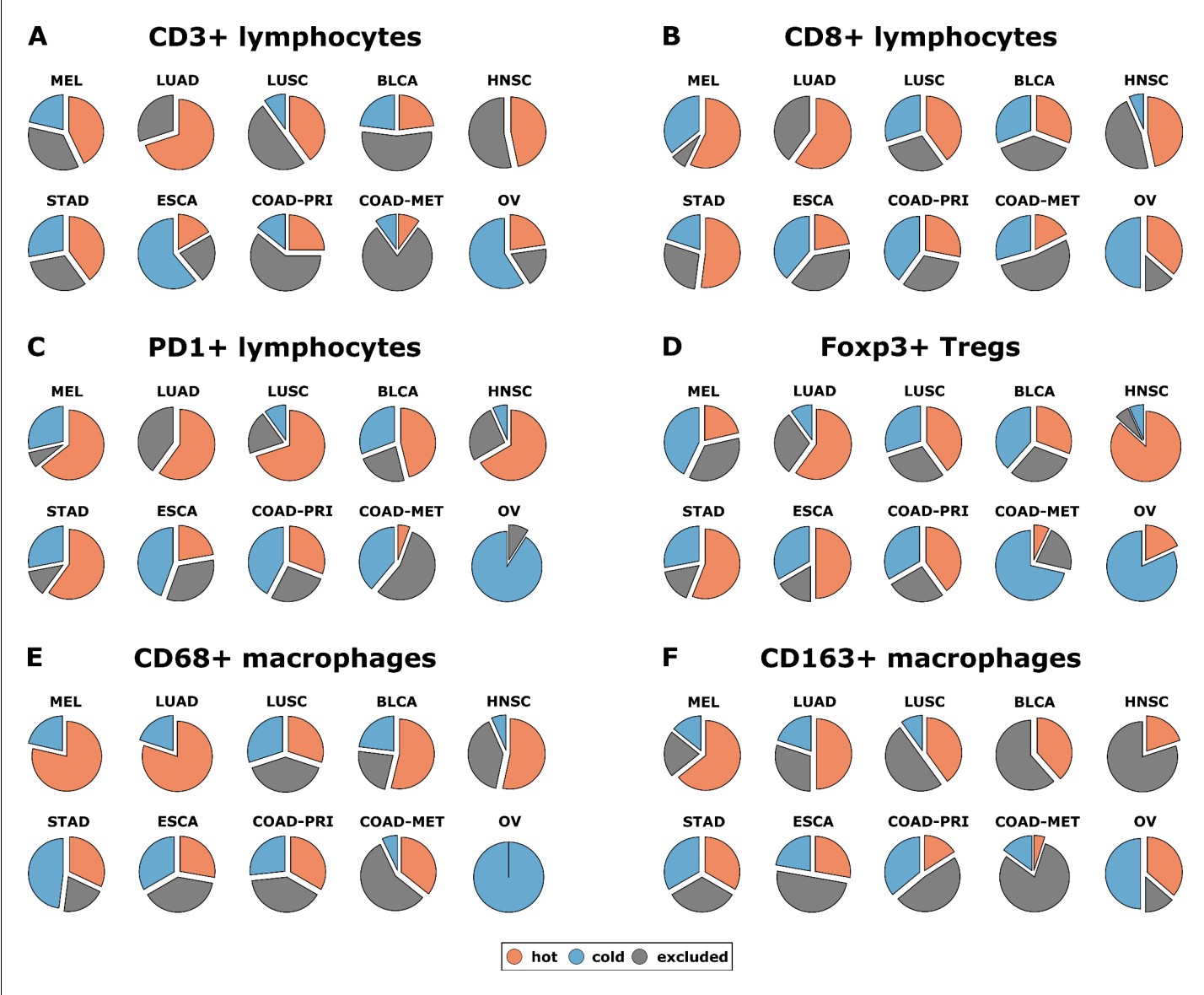

**Figure 3.** Distribution of immune topography phenotypes among different tumor types in the pan-cancer cohort. Analysis for all six immune cell types (A–F) and for all analyzed tumor types (MEL = melanoma, LUAD = lung adeno, LUSC = lung squamous, BLCA = bladder, HNSC = head and neck squamous, STAD = stomach adeno, ESCA = esophageal squamous, COAD-PRI = colorectal primary, COAD-MET = colorectal liver metastasis, OV = ovarian). These data comprise all N = 965 tissue slides from N = 177 patients. MEL through HNSC are to some degree sensitive to approved immunotherapies and predominantly have 'hot' phenotypes for most immune cells. However, among these tumor types, different phenotypes for immunosuppressive immune cells (Foxp3+ regulatory T-cells [Tregs]) and CD163+ macrophages prevail.
DOI: https://doi.org/10.7554/eLife.36967.010

immune excluded for these two cell types. The bivariate analysis showed that 'CD8-cold, CD163-cold' was the most prevalent category (*Figure 7D*).

We then used multivariable Cox proportional hazard models (including tumor stage, age and sex as potential confounders) to analyze the association between cell counts and cell topographies with survival. Cell densities of CD8 and CD163 in the tumor core and the outer invasive margin were not significantly correlated to overall survival (hazard ratios [HR] for death from any cause were 1.00 for both cell types in both compartments, *Supplementary file 5*). However, bivariate immune topographies showed significant association to overall survival: With 'CD8-cold, CD163-cold' as a reference

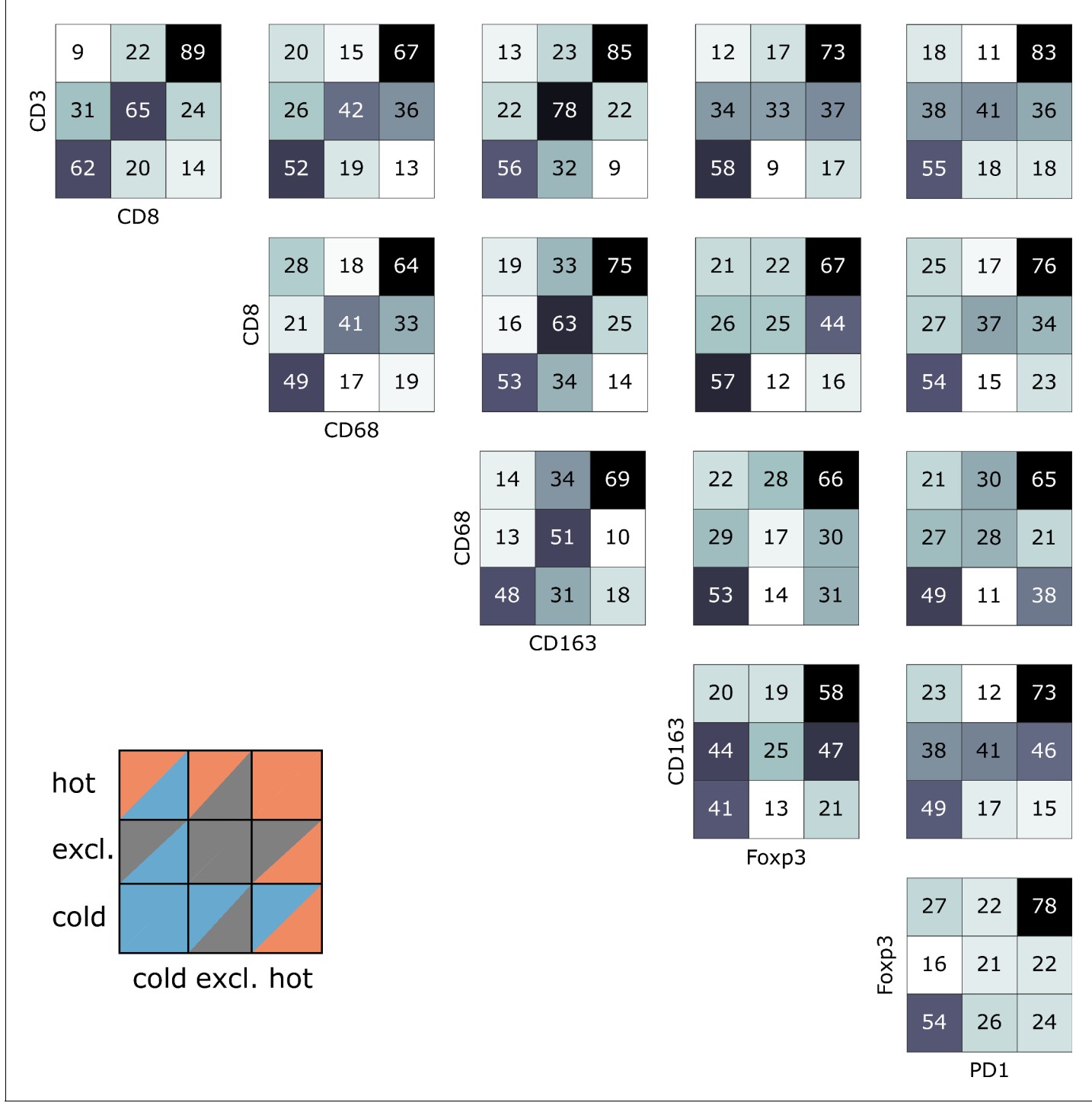

**Figure 4.** Pairwise analysis of immune phenotypes for all immune cell types in the pan-cancer cohort. For all tissue samples in all tumor types, a pairwise classification into cold-excluded-hot was done for all immune cell types. This analysis was based on the median cutoff for high and low cell densities. Absolute numbers of tumor are given, black marks the most abundant and white the least abundant group. For some pairwise comparisons such as CD3 and CD8 cells (top left corner), there is a high concordance between the phenotypes. For other comparisons such as Foxp3 and CD163 cells, phenotypes are mostly discordant.

DOI: https://doi.org/10.7554/eLife.36967.011

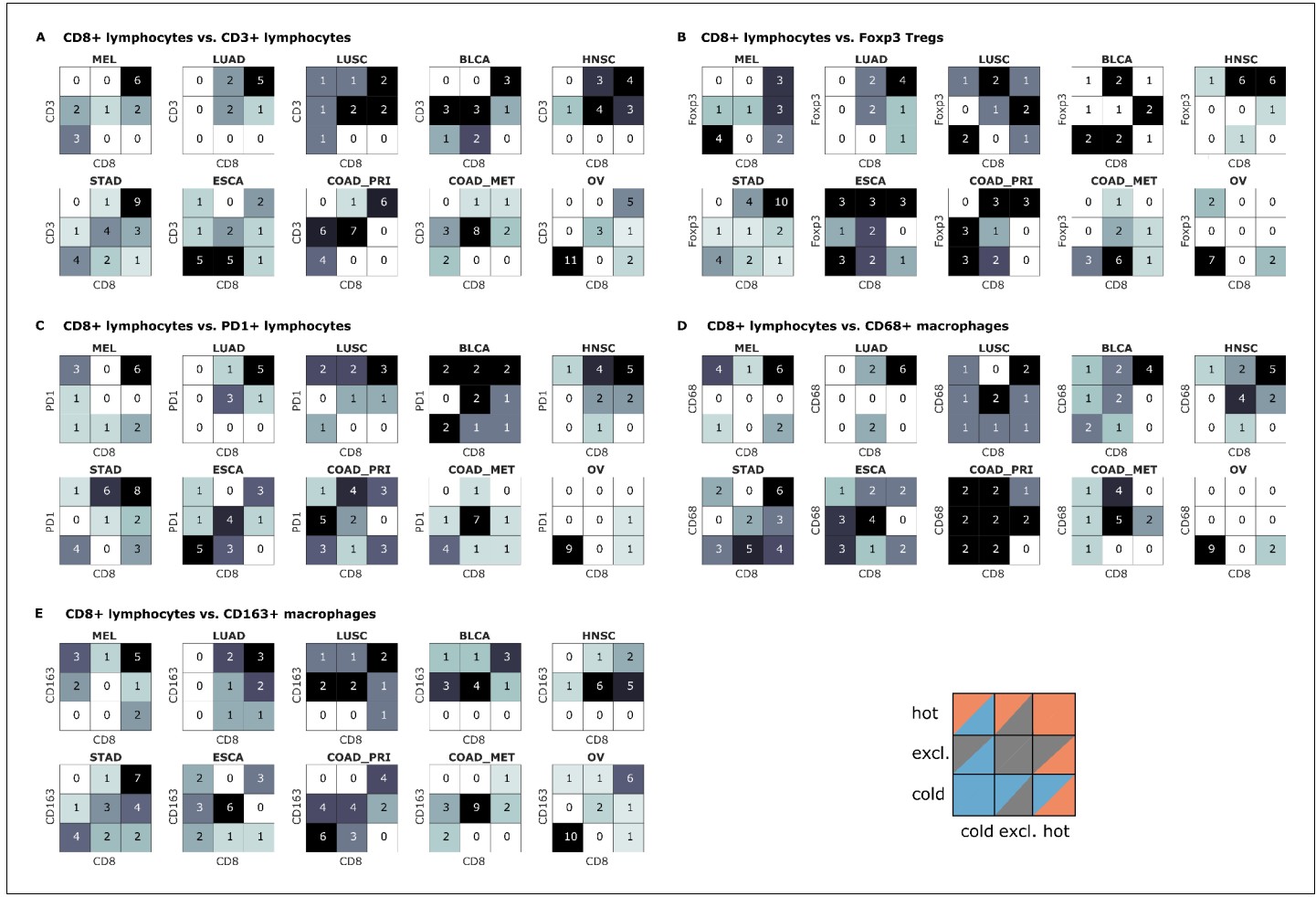

**Figure 5.** Bivariate immune phenotypes for each tumor type in the pan-cancer cohort. We analyzed the concordance between hot-cold-excluded topographies for CD8+ lymphocytes and all other cell types for each tumor type separately. Absolute numbers of patients assigned to each of nine phenotypes are overlaid on the heat map. For some cell types and some tumor types, immune topographies are concordant. This suggests that in these settings, a detailed analysis of multiple immune cell types in biomarker studies is not necessary. On the other hand, some cell types in other tumor entities (such as CD8+ lymphocytes and CD68+ macrophages in panel (D) show a high discordance in multiple tumor types. This suggests that measuring one of these cell types only may not be informative enough for biomarker studies.

DOI: https://doi.org/10.7554/eLife.36967.012

cohort, the HR was 1.75 for 'CD8-excluded, CD163 excluded' (p=0.041) and the HR was 2.71 for 'CD8-excluded, CD163-hot' (p=0.025, *Supplementary file 6* and *Figure 7D*).

## Discussion

The tumor microenvironment is a highly complex, heterocellular ecosystem (*Bindea et al., 2013*; *Tape, 2016*). Multiple immune cell types are involved in this system and ultimately shift it towards a tumor-promoting or tumor-rejecting environment (*Chen and Mellman, 2017*; *Fridman et al., 2017*). Also, these immune cells determine whether a tumor will be responsive to immunotherapy (*Becht et al., 2016a*). However, immunotherapy outcomes vary pronouncedly between different tumor entities and also between different patients within a given entity. The biological basis for these differences has been the subject of various studies but is still not entirely clear. Most importantly, we currently lack a comprehensive classification system for multiple effector parts of the immune system.

In the present study, we have performed a large-scale systematic analysis of lymphoid and myeloid phenotypes of human solid tumors. We provide evidence for a classification of tumor-immune

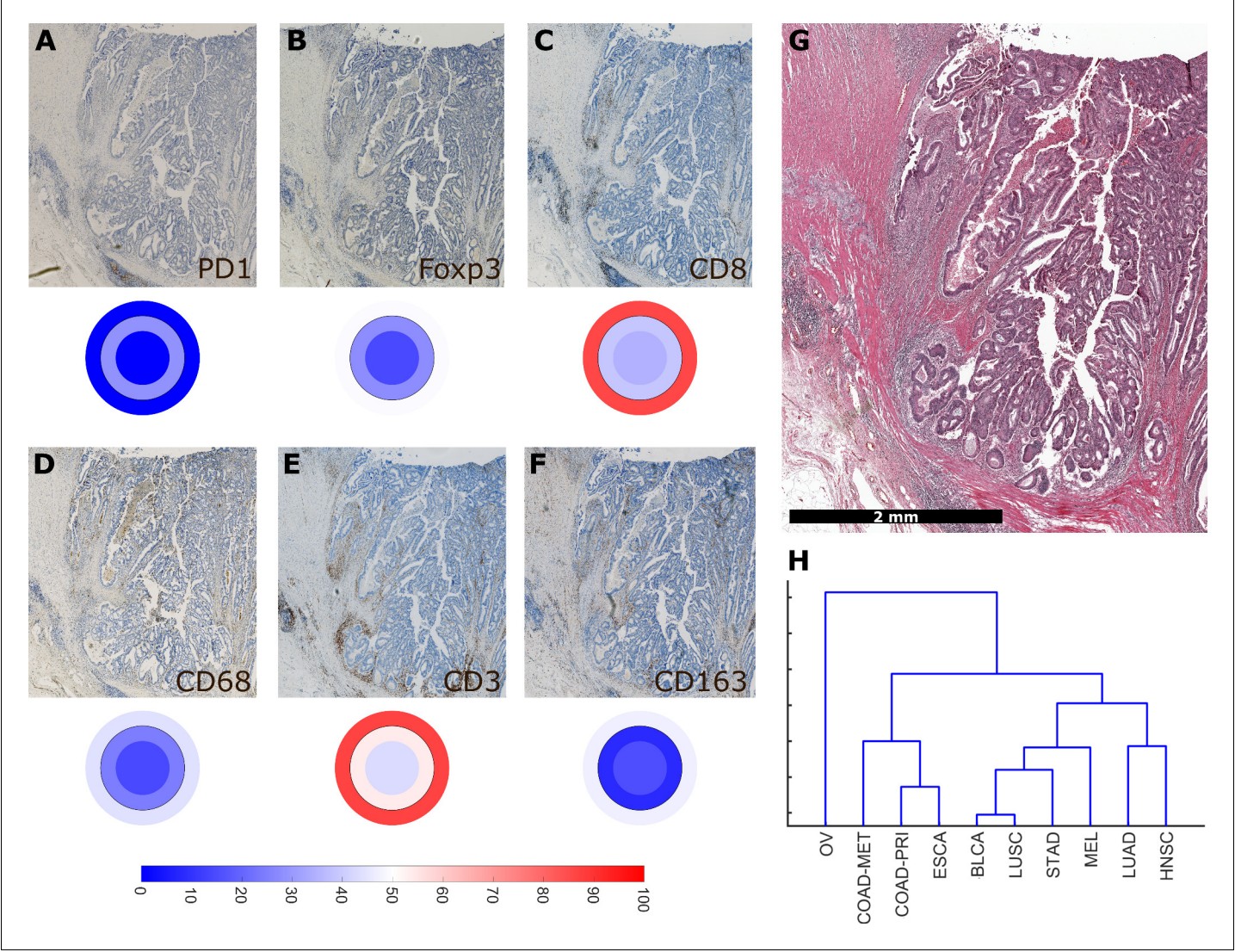

**Figure 6.** Overall similarity between tumor entities based on full immune topography. Hierarchical clustering based on all normalized cell densities of (A) PD1+exhausted lymphocytes; (B) Foxp3+regulatory T cells; (C) CD8+cytotoxic T lymphocytes; (D) CD68+monocytes/macrophages; (E) CD3 +lymphocytes and (F) CD163+pro tumor macrophages. Unit on the color scale: percentile-normalized cell density. (G) corresponding H & E image of the colorectal cancer sample used in this figure. (H) Hierarchical clustering of tumor types (N = 144 total patient samples).
DOI: https://doi.org/10.7554/eLife.36967.013

phenotypes into hot, cold and immune excluded spatial patterns. These patterns can be detected not only in lymphocyte infiltrates (lymphoid classification, as previously assumed by most studies) but also in macrophage infiltrates (myeloid classification). Interestingly, lymphoid and myeloid patterns are not always identical and a two-dimensional classification is needed to accommodate all possible lymphoid-myeloid phenotypes.

Addressing the biological differences and the different responses to immunotherapy across tumor entities, we systematically compared different tumor types. We analyzed ten tumor entities in the framework of this classification and showed pronounced differences, but also unexpected cross-entity similarities: We found that lymphoid-hot tumors are more prevalent in immunotherapy-responsive tumor entities such as melanoma and lung adenocarcinoma than in other entities such as colorectal cancer. Immune-excluded tumors, such as lymphoid-excluded and myeloid-excluded tumors are common in head and neck tumors (HNSC). Lastly, we show that characteristic patterns of

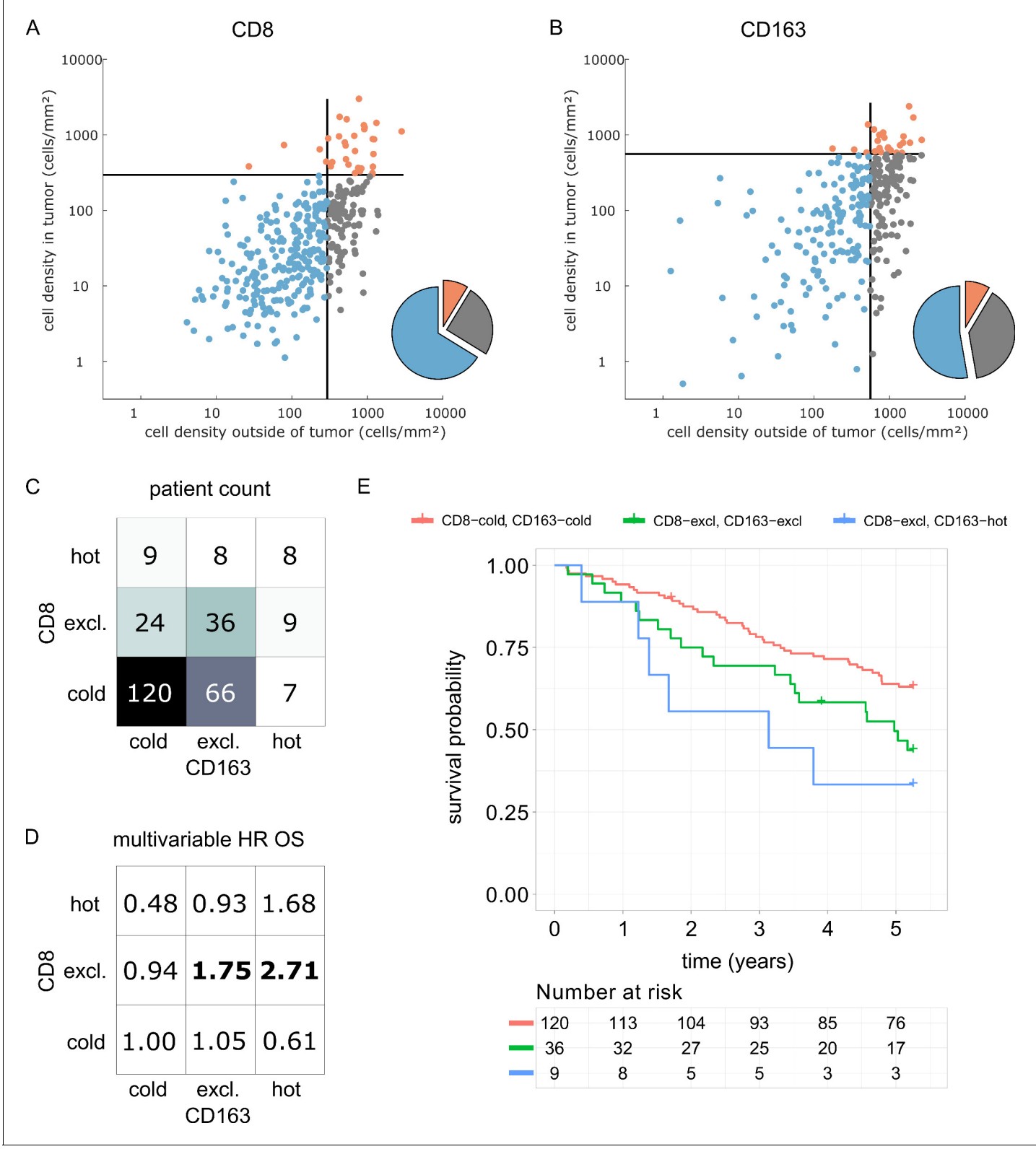

**Figure 7.** Prognostic value of the myeloid-lymphoid topography in primary colorectal cancer (CRC) in the DACHS cohort. In a validation cohort of N = 287 colorectal cancer patients (N = 286 with follow-up data) from the DACHS study, CD8 and CD163 staining of the primary surgical sample was correlated to clinical outcome (overall survival) using the cutoffs from the pan-cancer cohort. (**A**) As in the CRC subgroup in the pan-cancer cohort, CD8-cold was the most prevalent phenotype, followed by CD8-excluded. (**B**) A similar distribution of phenotypes was observed for CD163+ macrophages.
*Figure 7 continued on next page*

*Figure 7 continued*

(C) In the bivariate analysis for CD8 and CD163, most patients had cold and excluded phenotypes for both antigens. (D) For each of these nine phenotypes, the hazard ratio (HR) for death of any cause (inverse overall survival, OS) was derived from a multivariable Cox proportional hazard model (covariates UICC stage, age and sex). Bold HR indicates statistically significant findings with p<0.05. Bivariate analysis of both antigens is essential for risk stratification as CD8 and CD163 phenotypes show non-trivial interaction. Raw data for panel D are shown in *Supplementary file 5*. (E) The Kaplan-Meier plots for the reference groups (CD8-cold CD163 cold) and the two significant groups ('CD8-excl., CD163 excl' and 'CD8 excl., CD163 hot') show significant differences in overall survival. These differences are not captured by univariate, but only by this bivariate stratification system. Log rank p-value=0.019.

DOI: https://doi.org/10.7554/eLife.36967.014

immune evasion (via T cell exhaustion or infiltration by Tregs) are not exclusive to specific tumor-immune phenotypes but can be detected across the defined categories.

As part of this study, we present a clinical validation of the proposed classification system in the context of colorectal cancer (COAD). It has been previously shown that high density of CD8+ cells in COAD is associated with long survival (*Galon et al., 2006*) while high density of CD163+ cells is associated with poor survival (*Fridman et al., 2012*). However, some large studies have shown only a very modest impact of CD8+cell density on overall survival (*Glaire et al., 2018*). Likewise, in the DACHS cohort which we analyzed as part of this study, CD8+ cell density alone and CD163+ cell density alone were not clearly associated with differences in overall survival. Also, for both cell types, the immune topographies (cold, excluded, hot) were not associated with such differences. However, classifying patients according to bivariate (CD8 combined with CD163) immune topographies identified statistically significant associations to survival in a multivariable statistical model (*Figure 7D*). This was particularly clear for CD8-excluded tumors: in this case, CD163-cold, excluded and hot tumors had a significantly (log rank p-value 0.019) different prognosis (*Figure 7E*). These findings suggest that a multivariate analysis of spatial distributions of multiple immune cell types may be superior to merely quantifying a particular type of immune cells in COAD and possibly other human solid tumors.

Our study adds a systematic approach to a hitherto subjective classification of tumor-immune phenotypes. This new classification could constitute a novel framework to investigate immunotherapy responsiveness in clinical trials. Also, this classification sheds light on common immunological aspects by describing a shared immune topography across human solid tumors.

# Materials and methods

## Key resources table

| Reagent type (species) or resource | Designation | Source/Reference | Identifier | Additional information |
|---|---|---|---|---|
| Software, algorithm | QuPath v0.1.2 | Bankhead et al. | DOI: 10.1038/s41598-017-17204-5 | - |
| Antibody | Anti-human CD3 | Leica Novocastra | RRID:AB_563544 | Dilution 1:100 |
| Antibody | Anti-human CD8 | Leica Novocastra | RRID:AB_442068 | Dilution 1:50 |
| Antibody | Anti-human Foxp3 | eBioscience | RRID:AB_467555 | Dilution 1:100 |
| Antibody | Anti-human CD163 | BioRad | RRID:AB_2074540 | Dilution 1:500 |
| Antibody | Anti-human CD68 | Thermo Fisher Scientific | RRID:AB_720547 | Dilution 1:2000 |
| Antibody | Anti-human PD1 | Abcam | RRID:AB_881954 | Dilution 1:50 |

## Ethics statement and tissue samples

All experiments were conducted in accordance with the Declaration of Helsinki, the International Ethical Guidelines for Biomedical Research Involving Human Subjects (CIOMS), the Belmont Report

and the U.S. Common Rule. Anonymized archival tissue samples were retrieved from the tissue bank of the National Center for Tumor diseases (NCT, Heidelberg, Germany) in accordance with the regulations of the tissue bank and the approval of the ethics committee of Heidelberg University (tissue bank decision numbers 2152 and 2154, granted to NH and JNK, ovarian cancer tissues granted to SS; informed consent was obtained from all patients as part of the NCT tissue bank protocol, ethics board approval S-207/2005, renewed on 20 Dec 2017). Another set of tissue samples was provided by the pathology archive at UMM (University Medical Center Mannheim, Heidelberg University, Mannheim, Germany) after approval by the institutional ethics board (Ethics Board II at University Medical Center Mannheim, decision number 2017 – 806R-MA, granted to AM and waiving the need for informed consent for this retrospective and fully anonymized analysis of archival samples). Also, a set of melanoma samples was provided by the pathology archive at UMM after approval by the institutional ethics board (Ethics Board II at University Medical Center Mannheim, decision number 2014 – 835R-MA, granted to JU and waiving the need for informed consent for this retrospective and fully anonymized analysis of archival samples).

We analyzed tissue samples of primary esophageal carcinoma (ESCA), primary gastric cancer (STAD), primary colorectal cancer (COAD-PRI), primary lung adenocarcinoma (LUAD), primary lung squamous cell carcinoma (LUSC), primary head and neck squamous cell carcinoma (HNSC), primary bladder cancer (BLAC), ovarian cancer primary tumors (OV) as well as melanoma primary tumors (MEL) and colorectal cancer liver metastases (COAD-MET).

In addition to this pan-cancer cohort, we acquired a set of N = 287 primary surgical specimen of colorectal adenocarcinoma from the DACHS study (*Hoffmeister et al., 2015*; *Brenner et al., 2011*) which were provided by the NCT biobank under the same ethics board approval as stated above and including informed consent by all patients. Clinical data for this cohort are listed in *Supplementary file 1*.

## Immunohistochemistry

We performed histological staining for CD3 (dilution 1:100 with Leica antigen retrieval ER1 solution), CD8 (1:50 with ER1), Foxp3 (1:100 with Leica antigen retrieval ER2 solution), CD163 (1:500 with ER2), CD68 (1:2000 with Leica Fast Enzyme digestion) and PD1 (1:50 with ER1) on a Leica Bond automatic staining device using a hematoxylin-diaminobenzidine (DAB) staining protocol as described previously (*Halama et al., 2016*). For melanoma, FastRed (Leica #DS9390) was used as a chromogen. Stained whole slide tissue sections were digitized as described previously (*Halama et al., 2016*). Almost all samples were stained for these six immune cell markers. In cases of insufficient tissue availability, only CD3, CD8 and CD163 staining was performed.

## Image analysis

Our image analysis pipeline was composed of several steps: First, manual annotation of three regions of interest (ROI) in each histological whole slide image (*Figure 1A*). The ROIs were 'tumor core' (TU_CORE), 'inner invasive margin' (MARG_500_IN) and 'outer invasive margin' (MARG_500_-OUT). Invasive margins were 500 µm wide. Second, we automatically counted all positively stained cells using the open source software QuPath (*Figure 1B*) (*Bankhead et al., 2017*). Intensity thresholds and other parameters for cell detection and classification were set manually for each staining type and were identical for all samples in the pan-cancer cohort. All parameters are listed in *Supplementary file 2*. All cell detection scripts were manually checked for plausibility in all tumor entities. Examples for cell detection are shown in *Figure 1—figure supplement 1*. For all further analyses, cell density values were normalized by percentile within each staining type and were visualized as 'target plots' (*Figure 1C*). Staining intensity thresholds were slightly adapted for the DACHS cohort as listed in *Supplementary file 2*.

## Reproducibility

To ensure reproducibility of our digital pathology pipeline, we randomly selected 60 tissue specimen and repeated all analysis steps. We used new tissue slides with a distance between 4 µm and 40 µm from the original slice. We stained 30 of these slides for a macrophage marker (CD163) and 30 slides for a lymphocyte marker (CD3) and a blinded observer delineated ROIs for cell quantification as

before. There was a high correlation (Pearson's correlation coefficient was >0.74, p-value<0.001) between these replicates (*Figure 2—figure supplement 1*).

Furthermore, two slides that served as negative controls for CD3 and CD163 staining, 231/86516 (<<0.1%) and 1/38311 (<<0.1%) cells were false positive.

## Automatic determination of optimal cluster number

Analysis of all six immunostains for 177 patients in our collective yielded 965 sets of cell density counts in three spatial compartments each (2895 data points in total, missing values due to tissue availability or quality). For visualization, cell densities were subjected to percentile normalization (quantile normalization with 100 quantiles) for each staining type, across all tumor entities. For clustering and all other analyses, we used absolute cell density (cell number per $mm^2$).

To determine the optimal number of clusters in this data set, we used three different methods for clustering: a gaussian mixture model, k-means and hierarchical clustering. For 1 up to 12 clusters, we computed three loss functions for each approach, using the Davies-Bouldin (*Davies and Bouldin, 1979*), the Calinski-Harabasz (*CalińskiCalinski and Harabasz, 1974*) or the silhouette (*Rousseeuw, 1987*) criteria for quality of clustering. For all optimization methods, we performed 10 technical replicates.

## Implementation and data availability

All image analysis steps were implemented in QuPath (see key resources) and all downstream analyses were implemented in MATLAB (Mathworks, Natick, MA, USA) R2017a. All experiments were run on a standard workstation (Intel i7 Processor, 8 cores, 32 GB RAM, Microsoft Windows 10.1). We release all source codes under an open access license (*Kather, 2018*; copy archived at https://github.com/elifesciences-publications/immuneTopography). Also, we release all raw data from our experiments (*Supplementary file 3*). All survival analyses were performed in R version 3.5.1 (R-project.org) using the packages survminer, survival, ggfortify and ggplot2.

# Acknowledgements

The authors would like to thank Rosa Eurich and Jana Wolf (National Center for Tumor Diseases, Heidelberg, Germany), Katrin Wolk (University Medical Center Mannheim, Mannheim, Germany) and Nina Wilhelm (NCT Biobank, National Center for Tumor diseases, Heidelberg, Germany) for expert technical assistance. The authors are grateful to the participants of the DACHS study, the cooperating clinics which recruited patients for this study, and the Institute of Pathology, University of Heidelberg, for providing tissue samples for this study.

# Additional information

### Funding

| Funder | Author |
|--------|--------|
| Heidelberg School of Oncology | Jakob Nikolas Kather |
| Bundesministerium für Bildung und Forschung | Alexander Marx |

The funders had no role in study design, data collection and interpretation, or the decision to submit the work for publication.

### Author contributions

Jakob Nikolas Kather, Conceptualization, Data curation, Formal analysis, Validation, Investigation, Visualization, Methodology, Writing—original draft; Meggy Suarez-Carmona, Conceptualization, Resources, Formal analysis, Writing—review and editing; Pornpimol Charoentong, Conceptualization, Formal analysis, Writing—review and editing; Cleo-Aron Weis, Timo Gaiser, Conceptualization, Resources, Methodology, Writing—review and editing; Daniela Hirsch, Resources, Writing—review and editing; Peter Bankhead, Conceptualization, Software, Methodology, Writing—review and

editing; Marcel Horning, Data curation, Formal analysis, Methodology, Writing—review and editing; Dyke Ferber, Conceptualization, Methodology, Writing—review and editing; Ivan Kel, Conceptualization, Validation, Visualization, Methodology, Writing—review and editing; Esther Herpel, Resources, Methodology, Project administration, Writing—review and editing; Sarah Schott, Conceptualization, Resources, Data curation, Writing—review and editing; Inka Zörnig, Conceptualization, Data curation, Supervision, Project administration, Writing—review and editing; Jochen Utikal, Resources, Data curation, Investigation, Project administration, Writing—review and editing; Alexander Marx, Conceptualization, Resources, Formal analysis, Funding acquisition, Writing—review and editing; Herrmann Brenner, Resources, Data curation, Funding acquisition, Project administration, Writing—review and editing; Jenny Chang-Claude, Resources, Formal analysis, Funding acquisition, Investigation, Methodology, Writing—review and editing; Michael Hoffmeister, Resources, Formal analysis, Supervision, Funding acquisition, Writing—review and editing; Dirk Jäger, Conceptualization, Supervision, Project administration, Writing—review and editing; Niels Halama, Conceptualization, Supervision, Validation, Writing—review and editing

## Author ORCIDs

Jakob Nikolas Kather http://orcid.org/0000-0002-3730-5348
Peter Bankhead http://orcid.org/0000-0003-4851-8813
Marcel Horning http://orcid.org/0000-0003-1468-4645
Ivan Kel http://orcid.org/0000-0002-1221-7114

## Ethics

Human subjects: All experiments were conducted in accordance with the Declaration of Helsinki, the International Ethical Guidelines for Biomedical Research Involving Human Subjects (CIOMS), the Belmont Report and the U.S. Common Rule. Anonymized archival tissue samples were retrieved from the tissue bank of the National Center for Tumor diseases (NCT, Heidelberg, Germany) in accordance with the regulations of the tissue bank and the approval of the ethics committee of Heidelberg University (tissue bank decision numbers 2152 and 2154, granted to NH and JNK, ovarian cancer tissues granted to SS; informed consent was obtained from the patients as part of the NCT tissue bank protocol). Another set of tissue samples was provided by the pathology archive at UMM (University Medical Center Mannheim, Heidelberg University, Mannheim, Germany) after approval by the institutional ethics board (Ethics Board II at University Medical Center Mannheim, decision number 2017-806R-MA, granted to AM and waiving the need for informed consent for this retrospective and fully anonymized analysis of archival samples). Also, a set of melanoma samples was provided by the pathology archive at UMM after approval by the institutional ethics board (Ethics Board II at University Medical Center Mannheim, decision number 2014-835R-MA, granted to JU and waiving the need for informed consent for this retrospective and fully anonymized analysis of archival samples). In addition to this pan-cancer cohort, we acquired a set of N=287 primary surgical specimen of colorectal adenocarcinoma from the DACHS study which were provided by the NCT biobank under the same ethics board approval as stated above and including informed consent by all patients.

## Decision letter and Author response

Decision letter https://doi.org/10.7554/eLife.36967.023
Author response https://doi.org/10.7554/eLife.36967.024

# Additional files

## Supplementary files

• Supplementary file 1. Clinical characterization of the DACHS cohort. This table lists summary statistics of all relevant clinico-pathological features of the DACHS cohort.
DOI: https://doi.org/10.7554/eLife.36967.015

• Supplementary file 2. List of all image analysis parameters. In this table, all parameters for cell detection and classification using the open source software QuPath are listed. Two sets of

parameters are distinguished: 'DAB' (diaminobenzidine), used for blue-brown staining, and 'Red', used for blue-red staining in melanoma. OD = optical density. All parameters were used in the pan-cancer cohort unless labeled as 'DACHS', in which case they were used in the DACHS cohort.
DOI: https://doi.org/10.7554/eLife.36967.016

• Supplementary file 3. List of all samples and all measurement values of the pan-cancer cohort. In this table, we report all raw measurements for all samples that were used in this study. Column names are: 'class' (tumor type as listed above), 'patient' (patient pseudonym), 'antigen' (antigen for immunostain), 'TU_CORE_cells_mm2' (number of positively stained cells per square millimeter in the tumor core), 'MARG_500_IN_cells_mm2' (number of positively stained cells per square millimeter in the inner invasive margin, defined as ranging 0–500 μm to the inside from the tumor edge), 'MARG_500_OUT_cells_mm2' (number of positively stained cells per square millimeter in the inner invasive margin, defined as ranging 0–500 μm to the outside from the tumor edge).
DOI: https://doi.org/10.7554/eLife.36967.017

• Supplementary file 4. List of all cutoff values for all cell types. On the full data set of N = 965 tissue slides from N = 177 patients in 10 tumor types, we calculated the median cell density for each antigen, taking the compartments 'outer invasive margin' and 'tumor core' into account. These median values were subsequently used as cutoff values for low and high cell densities which were then used to define hot, cold and excluded phenotypes.
DOI: https://doi.org/10.7554/eLife.36967.018

• Supplementary file 5. Continuous cell densities of CD8+ and CD163+ cells are not significantly associated with overall survival in colorectal cancer. A multivariable Cox proportional hazard model was fitted to all variables listed in this table. N = 286 CRC patients in the DACHS cohort, number of events = 108, significance codes (sig): *<0.05, **<0.01, ***<0.001. HR = hazard ratio, UICC = Union internationale contre le cancer.
DOI: https://doi.org/10.7554/eLife.36967.019

• Supplementary file 6. Bivariate immune phenotype predicts risk of death of any cause. A multivariable Cox proportional hazard model was fitted to all variables listed in this table. N = 286 CRC patients in the DACHS cohort, number of events = 108, significance codes (sig): *<0.05, **<0.01, ***<0.001. HR = hazard ratio, UICC = Union internationale contre le cancer.
DOI: https://doi.org/10.7554/eLife.36967.020

• Transparent reporting form
DOI: https://doi.org/10.7554/eLife.36967.021

### Data availability

We release all source codes under an open access license (http://dx.doi.org/10.5281/zenodo.1407435; copy archived at https://github.com/elifesciences-publications/immuneTopography). Also, we release all raw data from our experiments (Supplementary File 3).

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
