## [Decision Letter]

Thank you for submitting your article "Topography of cancer-associated immune cells in human solid tumors" for consideration by *eLife*. Your article has been reviewed by three peer reviewers, including Ian Tannock as the Reviewing Editor and Reviewer #1, and the evaluation has been overseen by a Reviewing Editor and Jeffrey Settleman as the Senior Editor. The following individuals involved in review of your submission have agreed to reveal their identity: Roberto Salgado (Reviewer #2); Evan W Newell (Reviewer #3).

The reviewers have discussed the reviews with one another and the Reviewing Editor has drafted this decision to help you prepare a revised submission.

We are sending the original reviews, but the main points are:

There are major questions about reproducibility of the method. Your team should show:

(i) That using your current algorithm you have reasonable agreement in the results and classification when duplicate sections from the same patient's tumour are analyzed – a sample to show reproducibility is important.

(ii) The effect of specifying different cluster numbers in your algorithm – say 2 or 4 instead of 3. To what extent are the results determined by the algorithm used?

(iii) You should recognize that some of your presentation is highly technical and needs to be written in a way that will be understood by most readers of *eLife*.

Other more specific comments requiring your response are given in the individual reviews below.

*Reviewer #1*:

The authors classify 965 tumour samples (obtained from tumour banks) from 177 patients, and use digital analysis of sections stained for immunohistochemistry (IHC) with 6 immune markers to classify them on the basis of infiltration of immune cells (T-lymphocytes and macrophages). The major contribution is looking at immune cell infiltration within the core, at the periphery (and outside) different regions of tumours. Important missing information is any measure of the reproducibility of their IHC, automated machine learning approach and tumour classification when undertaken on duplicate specimens (adjacent sections). Also, the technical language of the text and figure legends will make it very hard for most readers of *eLife* to have a good understanding of what is being done and plotted.

The authors comment that tumours that are more likely to respond to immunotherapy (melanoma and lung cancer) are more likely to have immune cell infiltration than tumour types that tend not to respond, such as colorectal tumours, but this is a group correlation, not individual. The real test would of course be to compare the topography of immune cell infiltration with outcome of modern immunotherapy (anti-PD1 or – PDL1) in individual patients.

Reviewer #2:

This is an interesting study, with, as major novelty, the dichotomy between lymphoid and myeloid clusters in human cancer, defining herewith the three major entities "cold", "hot" and "excluded".

A major question arises on the clustering. When you cluster data points you need to tell the algorithm how many clusters. The authors start saying that there are three classifications, using this concept to cluster their data by running the clustering algorithm with k=3. Of course, their data will then end up in 3 clusters. The Authors could have used k=2 and would have ended up with two clusters and said only two types exist. Could have used k=5 and gotten 5 different clusters, etc.

What is missing is evidence that three clusters give a classification system that is the most informative of outcome. PAM50, as an example, ended up with 5 classes/clusters based on trying to group patients in 2,3,4,5,6,7,8 groups and seeing if the groupings made any sense biologically based on expression data from 50 genes. They picked 5 and was reasonable but other subclasses definitely exist.

So, how valid is their statement that only three types of immune-cell patterns exist?

Why wasn't clustering analysis done with k=2,3,4,5 and 3 shown to be optimal?

It may be suggested to compare data with number of clusters 2,3,4,5 and then show outcomes comparing the clusters. To improve the paper in its current form a survival analysis based on these defined classes might be important. Is OS data available for the samples that would allow for survival analysis comparing outcomes for each cluster k=2,3,4,5 and that 3 is most informative of outcome with maximum HR or lowest p-value? If there is no survival data available, they should include that in discussion and reason why the survival analysis wasn't done, stating that in future studies a comparison with survival is important so that other groups can make progress on their findings.

Reviewer #3:

In the study by Kather et al., the authors use single color IHC to comprehensively assess the immune infiltration of 10 different types of tumor tissue (9 cancer types). By analyzing each of their 177 patient samples in three compartments (core, inner and outer invasive margins) with 6 different cellular stains, they set out to assess the prevailing notion that there are three main patterns of lymphocytic invasion of tumors: "cold/immune desert", "immune-excluded" and "hot". As described, this paradigm, while widely accepted and discussed is lacking in quantitative evidence, making this an important and timely study. The approach taken here is clearly described and rational, which makes it a useful example of how standard IHC can be used to objectively assess tumor immune cell infiltration in a high dimensional way. Based on their analysis, the authors conclude that their data are consistent with the prevailing view (3 major clusters of lymphocyte infiltration profiles) and suggest a more complex view that also incorporates the profiles of myeloid cell (CD68) infiltration.

Although I agree with the overall conclusions made, and despite the rigor in data collection and preprocessing, I am a little bit disappointed in the rigor of the interpretation of the data. Based on density measurements of CD3 and CD8 cells, a tSNE plot is constructed to display the range T cell infiltration phenotypes and this is useful. However, based on this and k-means clustering (using k=3), the authors conclude that there are 3 clusters of immune infiltrate profiles. While this is reasonable and roughly fits, based on the tSNE representation, some attempt to rigorously assess a best-fit in terms of cluster numbers, or otherwise, would have been much appreciated. For the analysis of CD68 staining, this is even worse. Only k-means clustering (k=3) was used to conclude that 3 clusters exist. This is exactly what this algorithm will produce if 3 clusters are requested. No other evidence is provided that 3 clusters of profiles should be used.

All in all, I think this is an interesting and important study. However, I think the authors should better assess and/or qualify their statements about the validity of the 3-cluster paradigm. Although I agree that their data support this general notion and their analyses provide interesting comparative insights about these tumor types, alternatives are not formally addressed or disproven.

Comments:

Although important and nicely framed, the hypothesis that 3 clusters of lymphocytic infiltration patterns exists is not adequately tested. What would the data need to look like for the authors to conclude otherwise? Perhaps some sort of best-fit analysis could be used to assess the validity of the 3-cluster claim – for both lymphocytes and myeloid cells. tSNE plots should be shown for myeloid cell data.

The tSNE analysis performed is informative but incomplete. Could these plots be shown for myeloid cell parameters as well? It might also be useful to perform this analysis on the entire dataset (similar to that shown in Figure 5H – e.g., one dot per patient, colored by cancer type or colored by cluster definition from Figure 2). This comment highlights the value of the dataset the authors are providing: it is exciting that many different data analysis/visualization strategies could be used to evaluate these data.

It is not clear why the authors have provided the exaggeration of 10 for tSNE and used a non-default parameter for this. Does this value largely influence the outcome of their analysis? Did the authors also experiment with various perplexity values for their analysis?

For the tSNE analysis of this number of events, the stochasticity and perplexity value might be important factors. The authors should show several replicate tSNE runs to assess the consistency of the pattern shown. To justify which of these runs to shown in the main figure, the authors should choose the run with the best fit values provided by the tSNE algorithm. In other words, I am concerned that the conclusion that the tSNE visualization fits well with k-means (k=3) clustering could be a coincidence for this particular run and perplexity setting and the authors should assess the reproducibility of this analysis.

---

## [Author Response]

We are sending the original reviews, but the main points are:There are major questions about reproducibility of the method. Your team should show:(i) That using your current algorithm you have reasonable agreement in the results and classification when duplicate sections from the same patient's tumour are analyzed – a sample to show reproducibility is important.

We have repeated the full pipeline (staining, scanning and image analysis) for 60 slides (line 311-316, Figure 2—figure supplement 1). The resulting cell counts were highly (rho>0.74) correlated to the original analysis. Also, we included negative controls (subsection “Reproducibility” demonstrating a near-negligible fraction of falsely positively stained cells.

(ii) The effect of specifying different cluster numbers in your algorithm – say 2 or 4 instead of 3. To what extent are the results determined by the algorithm used?

In the revised manuscript, we have added results from multiple unsupervised methods to determine an optimal number of clusters K. These methods converged on K=3 in ~25% of runs, but mostly converged on K=2 (subsection “Unsupervised clustering of lymphocyte densities separates hot and cold tumors”, Figure 2—figure supplement 4). Thus, we conclude that the reviewer’s concerns are correct: the inherent structure of the data does not provide evidence for the existence of three clusters.

However, after dichotomizing cell counts at the median, we saw that the phenotypes “high cell density”, “low cell density” and “high density outside, low density inside” were equally common (Figure 2, subsection “Conceptual definition of hot, cold and immune excluded tumors”). We investigated whether this simple, qualitative definition of topographies translates into a clinically meaningful biomarker. We stained, annotated and analyzed tumor tissue from 287 colorectal cancer patients and saw that indeed, some topographies predict overall survival (more details below).

Thus, we conclude that the definition of hot/cold/excluded immune topographies is useful although there is no fully unbiased, hypothesis-free definition of these phenotypes.

(iii) You should recognize that some of your presentation is highly technical, and needs to be written in a way that will be understood by most readers of eLife.

We have addressed this issue by revising major parts of the manuscript, making it much more understandable. Also, we have fundamentally revised the figures to present the data more clearly.

Other more specific comments requiring your response are given in the individual reviews below.Reviewer #1:The authors classify 965 tumour samples (obtained from tumour banks) from 177 patients, and use digital analysis of sections stained for immunohistochemistry (IHC) with 6 immune markers to classify them on the basis of infiltration of immune cells (T-lymphocytes and macrophages). The major contribution is looking at immune cell infiltration within the core, at the periphery (and outside) different regions of tumours. Important missing information is any measure of the reproducibility of their IHC, automated machine learning approach and tumour classification when undertaken on duplicate specimens (adjacent sections).

We have performed these replication experiments as required (subsection “Image analysis”). Comparing the cell densities to the original results, we saw a very high concordance (Figure 2—figure supplement 1). Also, we have added negative controls (subsection “Reproducibility”).

Also, the technical language of the text and figure legends will make it very hard for most readers of eLife to have a good understanding of what is being done and plotted.

We have addressed this by re-phrasing large parts of the manuscript.

The authors comment that tumours that are more likely to respond to immunotherapy (melanoma and lung cancer) are more likely to have immune cell infiltration than tumour types that tend not to respond, such as colorectal tumours, but this is a group correlation, not individual. The real test would of course be to compare the topography of immune cell infiltration with outcome of modern immunotherapy (anti-PD1 or – PDL1) in individual patients.

The reviewer is right: We hypothesize that immune topographies might be predictive of immunotherapy response, but we show this only on a group level, not on the level of individual patients. Unfortunately, whole tumor specimen that include the tumor core and the invasive margin are usually not available for patients prior to immunotherapy.

However, in the revised manuscript, we have analyzed an additional patient cohort of N=287 colorectal cancer patients (subsection “Pan-cancer similarity based on spatial immune phenotype”). In this cohort, we saw that immune topographies carry prognostic information for overall survival.

Reviewer #2:This is an interesting study, with, as major novelty, the dichotomy between lymphoid and myeloid clusters in human cancer, defining herewith the three major entities "cold", "hot" and "excluded".

Thank you for this positive response.

A major question arises on the clustering. When you cluster data points you need to tell the algorithm how many clusters. The authors start saying that there are three classifications, using this concept to cluster their data by running the clustering algorithm with k=3. Of course, their data will then end up in 3 clusters. The Authors could have used k=2 and would have ended up with two clusters and said only two types exist. Could have used k=5 and gotten 5 different clusters, etc.What is missing is evidence that three clusters give a classification system that is the most informative of outcome. PAM50, as an example, ended up with 5 classes/clusters based on trying to group patients in 2,3,4,5,6,7,8 groups and seeing if the groupings made any sense biologically based on expression data from 50 genes. They picked 5 and was reasonable but other subclasses definitely exist.So, how valid is their statement that only three types of immune-cell patterns exist?Why wasn't clustering analysis done with k=2,3,4,5 and 3 shown to be optimal?It may be suggested to compare data with number of clusters 2,3,4,5 and then show outcomes comparing the clusters. To improve the paper in its current form a survival analysis based on these defined classes might be important. Is OS data available for the samples that would allow for survival analysis comparing outcomes for each cluster k=2,3,4,5 and that 3 is most informative of outcome with maximum HR or lowest p-value? If there is no survival data available, they should include that in discussion and reason why the survival analysis wasn't done, stating that in future studies a comparison with survival is important so that other groups can make progress on their findings.

Thank you for this detailed critique. We have done exactly as required and performed an unbiased optimization procedure to find the best number of clusters (Figure 2—figure supplement 4). Indeed, the inherent structure of the data does not provide strong evidence for the existence of three clusters (see above).

However, in the revised manuscript we show that the inner invasive margin and the tumor core are highly correlated. Thus, only two compartments have to be considered: inside and outside of the tumor. With this reduced data set, we show that a pragmatic definition of hot/cold/excluded tumors (subsection “Conceptual definition of hot, cold and immune excluded tumors”) can explain patterns observed in our pan-cancer cohort.

As suggested by the reviewer, we went on and asked whether the classification system into hot/cold/excluded tumors is useful in a clinical setting. To this end, we acquired a large cohort of human colorectal cancer patients, more than doubling the number of patients in our study. We found that indeed, this classification system can be used as a biomarker of potential clinical usefulness (subsection “Pan-cancer similarity based on spatial immune phenotype”).Reviewer #3:In the study by Kather et al., the authors use single color IHC to comprehensively assess the immune infiltration of 10 different types of tumor tissue (9 cancer types). By analyzing each of their 177 patient samples in three compartments (core, inner and outer invasive margins) with 6 different cellular stains, they set out to assess the prevailing notion that there are three main patterns of lymphocytic invasion of tumors: "cold/immune desert", "immune-excluded" and "hot". As described, this paradigm, while widely accepted and discussed is lacking in quantitative evidence, making this an important and timely study. The approach taken here is clearly described and rational, which makes it a useful example of how standard IHC can be used to objectively assess tumor immune cell infiltration in a high dimensional way. Based on their analysis, the authors conclude that their data are consistent with the prevailing view (3 major clusters of lymphocyte infiltration profiles) and suggest a more complex view that also incorporates the profiles of myeloid cell (CD68) infiltration.

Thank you for this concise summary.

Although I agree with the overall conclusions made, and despite the rigor in data collection and preprocessing, I am a little bit disappointed in the rigor of the interpretation of the data. Based on density measurements of CD3 and CD8 cells, a tSNE plot is constructed to display the range T cell infiltration phenotypes and this is useful. However, based on this and k-means clustering (using k=3), the authors conclude that there are 3 clusters of immune infiltrate profiles. While this is reasonable and roughly fits, based on the tSNE representation, some attempt to rigorously assess a best-fit in terms of cluster numbers, or otherwise, would have been much appreciated. For the analysis of CD68 staining, this is even worse. Only k-means clustering (k=3) was used to conclude that 3 clusters exist. This is exactly what this algorithm will produce if 3 clusters are requested. No other evidence is provided that 3 clusters of profiles should be used.

This comment is in line with the feedback by the other reviewers. As discussed above, we have thoroughly addressed this point in the revised manuscript, removing unclear visualizations.

All in all, I think this is an interesting and important study. However, I think the authors should better assess and/or qualify their statements about the validity of the 3-cluster paradigm. Although I agree that their data support this general notion and their analyses provide interesting comparative insights about these tumor types, alternatives are not formally addressed or disproven.

Thank you for this positive feedback. As discussed above, we have fundamentally revised the manuscript and addressed these points as required.

Comments:Although important and nicely framed, the hypothesis that 3 clusters of lymphocytic infiltration patterns exists is not adequately tested. What would the data need to look like for the authors to conclude otherwise? Perhaps some sort of best-fit analysis could be used to assess the validity of the 3-cluster claim – for both lymphocytes and myeloid cells. tSNE plots should be shown for myeloid cell data.

We have done exactly as suggested (Results section).

The tSNE analysis performed is informative but incomplete. Could these plots be shown for myeloid cell parameters as well? It might also be useful to perform this analysis on the entire dataset (similar to that shown in Figure 5H – e.g., one dot per patient, colored by cancer type or colored by cluster definition from Figure 2). This comment highlights the value of the dataset the authors are providing: it is exciting that many different data analysis/visualization strategies could be used to evaluate these data.

As discussed above, the tSNE plot was not an optimal choice to present the data. We have revised the figures (new Figure 2—figure supplement 5) to present the data in a more straightforward way.

It is not clear why the authors have provided the exaggeration of 10 for tSNE and used a non-default parameter for this. Does this value largely influence the outcome of their analysis? Did the authors also experiment with various perplexity values for their analysis?

See above, the tSNE plot has been removed.

For the tSNE analysis of this number of events, the stochasticity and perplexity value might be important factors. The authors should show several replicate tSNE runs to assess the consistency of the pattern shown. To justify which of these runs to shown in the main figure, the authors should choose the run with the best fit values provided by the tSNE algorithm. In other words, I am concerned that the conclusion that the tSNE visualization fits well with k-means (k=3) clustering could be a coincidence for this particular run and perplexity setting and the authors should assess the reproducibility of this analysis.

See above, the tSNE plot has been removed.